# Adaptive Responses as Mechanisms of Resistance to BRAF Inhibitors in Melanoma

**DOI:** 10.3390/cancers11081176

**Published:** 2019-08-14

**Authors:** Azad Saei, Pieter Johan Adam Eichhorn

**Affiliations:** 1Department of Medicine, Center for Molecular Medicine, Karolinska Institutet, 171 76 Stockholm, Sweden; 2Karolinska University Hospital Solna, 171 76 Stockholm, Sweden; 3School of Pharmacy and Biomedical Sciences, Curtin University, Perth 6845, Australia; 4Curtin Health Innovation Research Institute, Faculty of Health Sciences, Curtin University, Bentley, WA 6102, Australia; 5Cancer Science Institute of Singapore, National University of Singapore, Singapore 117599, Singapore; 6Department of Pharmacology, Yong Loo Lin School of Medicine, National University of Singapore, Singapore 117597, Singapore

**Keywords:** BRAF, melanoma, adaptive responses, resistance

## Abstract

The introduction of v-raf murine sarcoma viral oncogene homolog B (BRAF) inhibitors in melanoma patients with BRAF (V600E) mutations has demonstrated significant clinical benefits. However, rarely do tumours regress completely. Frequently, the reason for this is that therapies targeting specific oncogenic mutations induce a number of intrinsic compensatory mechanisms, also known as adaptive responses or feedback loops, that enhance the pro-survival and pro-proliferative capacity of a proportion of the original tumour population, thereby resulting in tumour progression. In this review we will summarize the known adaptive responses that limit BRAF mutant therapy and discuss potential novel combinatorial therapies to overcome resistance.

## 1. Introduction

Over 200 different mutations have been identified in the v-raf murine sarcoma viral oncogene homolog B (BRAF) gene in various tumours, including a highly prevalent valine-glutamine substitution in codon 600 [1]. This specific mutation has been shown to enhance the kinase activity of BRAF by 700-fold, resulting in the constitutive activation of the Mitogen-Activated Protein Kinase (MAPK) pathway [2]. In malignant melanoma, BRAF^V600E^ is the most common alteration, with approximately 50% of all melanoma patients harbouring this allele [1]. The discovery that oncogenic activating mutations in BRAF occur in malignant melanoma paved the way for the development of a number of specific inhibitors targeting mutant BRAF, including the small-molecule inhibitor vemurafenib. Vemurafenib received approval by the U.S. Food and Drug Administration in 2011 for the treatment of non-resectable BRAF mutant melanoma [3,4,5]. Unfortunately, despite pronounced reduction in tumour burden and significant increases in patient survival, most responses to BRAF inhibitors remain transient, as a result of primary or acquired resistance. The primary mechanisms of resistance are present in approximately 50% of patients, with 15% of patients displaying no significant change in tumour shrinkage, while 35% of patients display modest shrinkage but do not meet the required RECIST criteria for partial response [4,6]. These primary mechanisms of resistance may be caused by the presence of existing genetic lesions or changes in expression in various components of the MAPK pathway, which results in the preservation of Extracellular Signal-Regulated Kinase (ERK) signaling, following BRAF inhibition [7,8,9,10,11,12,13,14,15]. Furthermore, the activation of parallel pathways, such as the Phosphoinositide 3-kinase (PI3K) pathway, can also support B-RAF inhibitor resistance [16]. However, resistance can also be facilitated through a number of adaptive responses resulting in recovery of ERK signaling following BRAF inhibition. The effect of these feedback loops on MAPK re-activation, independent of acquired mutations, has been suggested to allow cells to persist in an adapted drug tolerant state, eventually permitting acquired resistance to take hold [17]. 

To accurately translate cues from the extracellular environment into appropriate transcriptional outputs, all pathways, including the MAPK pathway, are tightly regulated by a number of different compensatory mechanisms or feedback loops. Under normal physiological conditions, these positive or negative feedback loops exist to regulate overall cellular homeostasis. Upon stimulation, positive feedback loops will amplify the output of the signal, so that the amplitude or duration of the output may be increased. In contrast, negative feedback loops function to limit the overall output of the signal, thereby preventing strong input signals from triggering maximal pathway activation. Importantly, these feedback loops can be initiated or attenuated during both activation or inhibition of the pathway, such that the inhibition of the pathway may result in the loss of a feedback loop, resulting in the partial recovery of the output (Figure 1). In cancer, unless genetically altered, these intrinsic feedback loops remain intact, such that targeted inhibition of the MAPK pathway may result in the stimulation or relief of compensatory mechanisms to re-activate MAPK signalling. It is the unintended consequences of these evolutionary established feedback loops that limit overall tumour responses to BRAF inhibitors. In this review we will discuss some of the known feedback loops that limit BRAF mutant therapy and highlight potential novel combinatorial therapies to overcome intrinsic resistance.

## 2. MAPK Signalling Pathway 

MAPK signalling pathway is an evolutionary conserved pathway, which helps cells sense and respond to the extracellular signals by regulating a wide range of cellular responses, including mitosis, metabolism, motility, differentiation, and proliferation. As such, the MAPK pathway is one of the most well-studied pathways and has been well characterised [18,19,20]. 

MAPK signalling is activated principally by ligand binding to receptor tyrosine kinases (Figure 2A). The resulting dimerization of the receptors promotes receptor activation and auto-phosphorylation of Tyr residues on the intracellular domains of the receptors. The alteration in charges induced by phosphorylation on the intracellular receptor motifs serves to act as binding sites for proteins containing Src homology 2 or phosphotyrosine-binding domains. This, in turn, permits the recruitment of adapter proteins, such as Growth factor Receptor bound protein 2 (GRB2) to the receptor. Subsequently, the guanine nucleotide exchange factor (GEF) son of seven-less (SOS) binds to GRB2 and accelerates the activation of RAS, by exchanging the inactive guanosine-5^1^-diphosphate (GDP-RAS) to active guanosine-5^1^-triphosphate (GTP-RAS). This nucleotide exchange permits the direct interaction of RAS with its downstream effectors most notably the serine/threonine kinases ARAF, BRAF, and CRAF (also known as RAF1). Ras activation is subsequently attenuated by the GTPase-activating protein (GAP) neurofibromin 1 (NF1).

The family of RAF kinases constitute the most important downstream RAS effectors. The three RAF family members share three highly conserved regions denoted as CR1, CR2, and CR3 [21,22]. CR1 contains the Ras-binding domain and a cysteine-rich domain, which stabilize the inactive confirmation. CR2, is rich in serine and threonine phosphorylation sites, which are required for RAF membrane recruitment and activation, while CR3, contains the kinase domain [23]. Specifically, within the N-terminal of CR3, resides a P-loop structure required for ATP binding and maintaining BRAF in an inactive confirmation. Upon activation, RAS-GTP binds the RAS-binding domain in CR1 permitting phosphorylation at T599 and S602, within the P-loop of CR3. The resulting phosphorylated residues destabilize the interactions within the P-loop, permitting the activation segment to vault into an active confirmation and permit dimerization (reviewed in Karoulia et al. [24]). All RAF family members can dimerise with each other. However, dimerization preferences remain poorly understood. Nevertheless, it has been noted that BRAF/CRAF dimers primarily dictate RAS-dependent signaling. Upon dimerization and subsequent activation, RAF phosphorylates the dual specificity kinase Mitogen-Activated Protein Kinase Kinase 1 (MEK1) on S218 and S222 in an intricate process that involves the KSR scaffold. These phospho-sites are located in the activation segment of MEK kinase and mediate the activation of this enzyme. In return, activated MEK kinase phosphorylates the ERK Kinase (MAPK), which is translocated into the nucleus upon activation [25]. Over a 150 ERK substrates have been identified to date, including a number of transcription factors, which process the overall MAPK/ERK dependent biological response. 

The MAPK signalling pathway is essential for melanomagenesis, as indicated by hyper-activation of this pathway in up to 70–90% of melanomas (see Figure 2B). Furthermore, oncogenic mutations, in a number of core components of this pathway, have been shown to support melanoma development. Unfortunately, the majority of these molecular lesions also prime intrinsic resistance to RAF inhibitors through the constitutive activation of downstream ERK signalling [3,26,27]. Therapeutic interventions, combining BRAF inhibitors with MEK inhibitors, have shown promising results highlighted by improved clinical outcomes and reduced toxicities. However, the development of drug resistance continues to remain the most common eventual outcome [28,29]. 

## 3. Inhibitory Feedback Phosphorylation by Downstream Kinases

As highlighted above, inhibitors targeting activated BRAF^V600E^ have proven to be clinically active in patients harbouring this mutation. However, the overall response rates are limited by primary or acquired resistance. Apart from a number of genetic alterations, which drive resistance, tumour regression can also be limited by a number of compensatory mechanisms [31]. These intrinsic adaptive responses, leading to resistance, can be divided into: A) non-transcriptional-based adaptive responses, resulting in the regulation of post-translational modifications of upstream kinases; and B) transcriptionally mediated adaptive responses. Importantly, the majority of these intrinsic mechanisms of resistance are highlighted by the reactivation of MAPK signaling, decreasing the overall sensitivity to compounds targeting this pathway and decreasing overall patient responses.

Over 20 years ago it was first reported that an RAF inhibitor ZM 336372 induced a unexpected increase in RAF kinase activity [32]. Now, termed the “RAF inhibitor paradox”, this area of research has been under intense investigation. A consequence a deeper understanding of RAF dimerization and activation has partially clarified this phenomenon (reviewed in [24]). Vemurafenib was developed as a specific BRAF^V600E^ inhibitor, and even though it binds and inhibits all RAF isoforms in vitro, vemurafenib only significantly limits cellular proliferation in cells and tumours harbouring BRAF^V600E^ mutations. Disappointingly, the treatment of cells expressing wild type BRAF, with vemurafenib, inadvertently enhances RAS-RAF binding, resulting in increased RAF dimer formation and downstream ERK activation. As such, it is now recommended that vemurafenib not be used in BRAF^V600E^ patients harbouring amplifier mutations (i.e., NRAS). Furthermore, as a consequence of these effects, a number of new paradox-breaking inhibitors have been developed and are currently being tested in clinical trials [24]. 

As indicated, the reactivation of the MAPK pathway, following vemurafenib treatment, can occur through a loss of negative feedback loops. Nearly all of the core components of the canonical MAPK pathway are regulated through feedback inhibition (Figure 3A). Following catalytic activation, ERK phosphorylates the Epidermal Growth Factor Receptor (EGFR) at T669, within the juxtamembrane region, shifting the monomer-dimer equilibrium to ligand bound monomers [33]. Therefore, ERK phosphorylation functions through a negative feedback loop to inhibit the dimerization capability of the EGF receptors, attenuating continued EGFR activation. 

Similarly, the guanine nucleotide exchange factor SOS is regulated through feedback inhibition. ERK-mediated phosphorylation of SOS results in the dissociation of the GRB2-SOS complex and inhibiting RAS activation [34,35]. RAS activation is further modified through feedback upregulation of neurofibromin RAS-GAP activity [36]. Therefore, RAS activation is downregulated through both the disruption of the GRB2-SOS complex and upregulation of NF1 GAP activity. Furthermore, it has been speculated that the loss of NF1 in melanoma would result in prolonged RAS activation, rather than enhanced RAS signalling, potentially shifting the activation of the pathway from a pro-proliferative to a pro-differentiation [36]. SPROUTY proteins are evolutionarily conserved regulators of the MAPK signalling, acting at multiple nodes along the pathway, coordinating appropriate cellular responses [37]. SPROUTY phosphorylation, following MAPK activation, has been shown to create a docking site for GRB2 inhibiting GRB2 function and downstream RAS activation [38]. Furthermore, SPROUTY proteins have been shown to bind wild type BRAF and potentially inhibiting RAF activation [39]. Interestingly, SPROUTY cannot bind mutant forms of BRAF, therefore, the specific role of SPROUTY in ERK-mediated adaptive responses, following the treatment of vemurafenib, remains undetermined. 

To limit continuous RAF activation, ERK phosphorylates RAF at a number of different Serine/Threonine residues (S151, T401, S750, T753) [40]. While S151 has been shown to inhibit RAS-RAF complex formation, the other three phosphorylation sites appear to contribute to RAF dimerization [40]. 

Upon activation, MEK1 and MEK2 phosphorylate a number of substrates, including ERK on specific threonine residues. However, to maintain signalling equilibrium in MAPK pathway activation, MEK1 can form a stable complex with MEK2. These heterodimers play a crucial role in fine-tuning ERK signalling. Interestingly, as part of the negative feedback loop, ERK phosphorylates MEK1 at T292, a residue specific to MEK1, decreasing dimerization and inhibiting further ERK activation [41]. If heterodimerization is prevented by decreased MEK1 expression, or through the introduction of a equivalent mutation in the dimer interface, ERK mediated feedback is lost. 

From the perspective of MAPK pathway inhibitor resistance, it is important to note that the reactivation of MAPK pathway alone, through the loss of these kinase-dependent negative feedback loops, is unlikely to be sufficient to drive resistance, but rather allow a sub-population of cells to exist in an adapted drug-tolerant state [17,42]. Nevertheless, it was in this final realisation, that the reactivation of the MAPK pathway, independent of acquired mutations, was inherently limiting overall survival. This has spurred the investigation into clinical trials, using combinatorial approaches targeting both BRAF and downstream MEK [3]. 

## 4. Transcriptionally-Mediated Feedback Loops Following MAPK Pathway Inhibition 

In addition to the regulation by downstream kinases, MAPK pathway activity can also be regulated by transcriptionally-mediated adaptive responses. However, unlike immediate direct phosphorylation by ERK, following pathway activation, ERK mediated transcriptional responses are likely to result in a delayed response to MAPK signalling. These protracted feedback loops are likely to be involved in prolonged MAPK signalling, rather than affecting the overall amplitude of ERK signalling (Figure 3B). Furthermore, changes in gene expression, following MAPK inhibition, has also been shown to influence Reactive Oxygen Species (ROS) production, epigenetic alterations, and autophagy [16,43]. 

### 4.1. DUSP 

The dual specificity MAPK phosphatases (DUSPs) are transcriptionally-induced upon long term activation of MAPK signalling and can principally remove phospho groups from phosphorylated residues on ERK kinase. Therefore, DUSPs are considered as the prototypical negative feedback loop, resulting in the downregulation of MAPK signalling [44]. Dual specificity MAPK phosphatases (DUSPs or MKPs) are the largest group of phosphatases known to dephosphorylate ERK1/2 kinase. MAPK phosphatases have similar structures, with non-catalytic N-terminal domain and C-terminal catalytic domains. The dual specificity of these enzymes refers to their ability to remove phospho-groups from both Threonine (T) and Tyrosine (Y) sites in the activation loop of ERK enzymes [45]. It is worth mentioning that ERK requires the phosphorylation of both threonine and tyrosine residues in its catalytic domain for full enzymatic activation. DUSPs are classified into three groups, based on their sub-cellular localization and specificity. The first group is made up of the nuclear DUSPs, including DUSP1, 2, 4, and 5. The second group is cytoplasmic, which includes DUSPs 6, 7, and 9. The last group includes DUSPs 8, 10, and 16. Among DUSP family members, DUSP5 specifically removes phospho- residues from ERK1/2 kinase, while DUSP8 is specific for the p38/JNK (c-Jun N-terminal kinase) pathway. Although several MAPKs signalling pathways, besides the ERK1/2 pathway, have been identified, here we focus only on the regulation of canonical ERK1/2 mediated MAPK signalling by DUSP enzymes. 

DUSP expression has been shown to be induced through Fibroblast Growth Factor treatment, resulting in the activation of ERK and subsequent binding of the ETS proto-oncogene 1 (ETS1) transcription factor to the DUSP6 promoter [46]. Interestingly, a recent study showed that melanoma cell lines and primary tissues show some of the highest expressions of DUSP6 among all cancer cell lines and primary tissues tested. Moreover, it has been shown that the gene expression of some of the DUSPs, such as DUSP4 and DUSP6 are under the direct control of oncogenic signaling. BRAF^V600E^, or RAS with BRAF and NRAS mutated cell lines, showed higher levels of DUSP6 expression, compared to melanoma cell lines carrying wild type BRAF and NRAS [47,48]. This activity suggests a clear relationship with MAPK pathway activation and DUSP6 expression in melanoma [49,50]. Therefore, DUSP6 is now used as a predictive biomarker for the clinical efficiency of MAPK inhibitors in melanoma [51]. Moreover, DUSP6 was recently identified as among the few genes where expression was rapidly downregulated, following their treatment with BRAF inhibitors (vemurafenib and RAF265) or MEK inhibitors (PD184352 and U0126) [48]. This loss of negative feedback inhibition has been shown to play a role in resistance to MAPK inhibitors.

Some of the other DUSP family members, including DUSP5, have been shown to, not only dephosphorylate and inactivate ERK, but also trap ERK in the nucleus, increasing the activity of MAPK signalling pathway in the cytoplasm [52]. This activating role of DUSPs occurs through blocking ERK-mediated BRAF inhibition. Inherently, the depletion of DUSP5 in cells harbouring BRAF^V600E^ mutations leads to oncogene induced senescence [52]. This recently identified oncogenic role of DUSPs on ERK pathway activation contrasts with the expected function of these enzymes as tumour suppressors that tend to oppose MAPK signalling. These results have surprisingly also highlighted the potential of using DUSP inhibitors to treat tumours with activating mutations in the MAPK pathway. Moreover, targeting DUSP6, using a specific DUSP6 inhibitor, named BCI, re-sensitized ovarian cancer cells to chemotherapy treatment by enhancing ERK mediated apoptosis. Specifically, it was shown that BCI treatment, in combination with Paclitaxel, led to the inhibition of DUSP6, and increased levels of phosphorylated ERK and EGR1 transcription factor, promoting apoptosis [53].

### 4.2. SOX10

The sex determining region Y-box 10 (*SOX10*) is a member of the SOX family of transcription factors, which play a crucial role in the development of the neural crest and melanocyte lineage. Notably, SOX10 haplo-insufficiency causes pigmentation defects and Waardenburg syndrome in humans. Recently, SOX10 has also been shown to regulate sensitivity to BRAF inhibitors in melanoma through compensatory mechanisms [9,54,55,56]. SOX10 regulates the proliferation and survival of melanocytes through the upregulation of a plethora of genes, including Microphathalemia-associated transcription factor (MITF), Dopachrome tautomerase (DCT), Tyrosinase (Tyr), the lncRNA SAMMSON, and the forkhead box D3 (FoxD3) transcription factor [55,56,57,58,59]. Forkhead box D3 (FOXD3) plays a role as a stemness-related transcription factor, maintaining the pluripotent stem cells, and blocking the generation of melanocyte progenitors from neural crest precursors during development. FOXD3 expression is induced by RAF and MEK inhibitors through SOX10, whereby the loss of ERK mediated phosphorylation of SOX10 permits the recruitment of SOX10 to the FOXD3 promoter [11,55]. This induction occurred specifically in BRAF mutant melanoma cells, while the FOXD3 induction was not observed in BRAF wild type melanoma, and mutant BRAF thyroid cancer cells, treated with MEK inhibitors [60]. In line with these results, the downregulation of SOX10 led to decreased levels of FOXD3 upon inhibition of MAPK pathway [55]. It is worth noting that knock-down of either, SOX10 or FOXD3, leads to G1 cell cycle arrest, upregulation of pro-apoptotic proteins Caspase 1 (CASP1), and increased levels of cyclin dependent kinases (CDK) inhibitors p21^cip1^ and p27^kip1^, while downregulating CDK protein levels, Retinoblastoma protein (RB), and anti-apoptotic proteins, such as Bcl-2. These changes collectively lead to senescence and unresponsiveness to BRAF inhibition [61,62]. 

One of the proposed mechanisms of FOXD3-mediated adaptive resistance, following MAPK inhibition, is the direct transcriptional upregulation of the RTK, Erb-B2 Receptor Tyrosine kinase 3 (ERBB3). The enhanced ERBB3 signalling, with concomitant upregulation of both downstream MAPK and PI3K pathways, would expectantly block the anti-proliferative and cytotoxic effects of MAPK inhibition. 

Another target gene of SOX10 is the long, non-coding RNA (lncRNA), SAMMSON. SAMMSON is expressed in 90% of human melanomas and is essential to the viability of melanomas, irrespective of BRAF or NRAS mutational status [63]. The p32 protein is an essential ingredient in the maturation of the mitochondrial 16S RNA and, therefore, has been directly implicated in the maintenance of mitochondrial membrane potential and oxidative phosphorylation (OXPHOS) [64]. SAMMSON interacts with p32 and enhances the enzymatic activity of the respiratory complexes required for these processes. Importantly, systematic targeting of SAMMSON, with anti-sense oligos, enhanced the anti-proliferative capacity of mutant BRAF inhibitors, without having an effect on overall melanocyte integrity, highlighting the potential for combination therapy targeting mechanisms, which regulate OXPHOS with BRAF inhibitors [63,65]. 

In contrast to the role of SOX10 as a biomarker for BRAF inhibitor resistance, the suppression of SOX10 has also been shown to enhance Transforming Growth Factor β (TGFβ) signalling, leading to the upregulation of platelet-derived Growth Factor Receptor-β (PDGFRβ) and EGFR [9]. The upregulation of these receptor tyrosine kinases are considered to be one of the crucial mechanisms for the adaptive resistance to BRAF and MEK inhibitors [9,66]. The contradictory results for SOX10, may be partially explained by the different temporal regulation of SOX10-mediated gene expression in immediate BRAF inhibitor treatment versus prolonged BRAF inhibitor treatment. SOX10-mediated FOXD3 upregulation occurs rapidly following BRAF inhibition; however, SOX10-mediated repression of TGFβ/EGFR/PDGFRβ occurs following the prolonged treatment with BRAF inhibitors, suggesting that differential drug-induced transcriptional states may switch SOX10 from an oncogene to a tumour repressor [66]. This may be insinuated as a treatment of melanoma cell lines with TGF-β ligand, or where the forced activation of EGFR has decreased the rate of proliferation and induced a phenotype similar to oncogene induced senescence [9]. However, this effect could be reversed following treatment with either, BRAF or MEK inhibitors. Consistently, the downregulation of SOX10 in untreated melanoma cells was effective in reducing tumour growth, but SOX10 depletion, in treated melanoma cells, led to resistance, and unresponsiveness, to vemurafenib [9,55]. Therefore, targeting SOX10 in combination with BRAF inhibitors and EGFR or ERBB3 inhibitors may be a promising therapeutic strategy in overcoming resistance to MAPK pathway inhibitors. 

### 4.3. MITF

Similar to SOX10, the Melanocyte Inducing Transcription Factor (MITF), plays a crucial role in the development and differentiation of melanocytes from the neural crest [67]. Studies analyzing biopsies, derived from melanoma patients and xenograft samples, have shown that MITF expression was commonly upregulated in response to MAPK inhibition [68,69]. Furthermore, it has been demonstrated that increased MITF expression renders melanoma cells resistant to BRAF pathway inhibition [70,71]. Melanocyte/melanoma specific isoform of MITF (M-MITF) is the only isoform of MITF where its expression is under the control of BRAF^V600E^. Although, it was shown that ERK kinase phosphorylates MITF at S73, a site targeting MITF for proteasomal degradation, other mechanisms were shown to be responsible for regulating mRNA expression of MITF [18]. One of these mechanisms, is the positive regulation of MITF by the receptor tyrosine kinase TYRO3, through a SOX10 dependent manner [72]. Downregulation of TYRO3 significantly inhibited the proliferation of melanoma cells, while the ectopic expression of TYRO3 inhibited BRAF^V600E^ induced senescence and enhanced cell proliferation [72]. 

The role of MITF as a mechanism of resistance to BRAF inhibition; however, was contradicted by findings, that show lower levels of MITF expression upon inhibition of MAPK signaling in melanoma, and a recent report, which predicts early resistance to ERK inhibitors in melanoma samples with low MITF and AXL receptor tyrosine kinase (AXL) ratios [73,74]. Furthermore, it has been shown that MITF expression is lost in BRAF^V600E^ mutated melanoma, and that a low expression of MITF leads to increased metastatic ability of melanoma cells [75]. Therefore, pushing the expression of MITF above endogenous levels has been suggested as a therapeutic opportunity to treat metastatic melanoma harboring the BRAF^V600E^ oncogene [76]. The enforced expression of MITF was demonstrated to induce the differentiation and lower cell proliferation in BRAF^V600E^ mutated melanoma cells, thereby highlighting the equivocal role of MITF in melanoma progression and BRAF inhibitor resistance. The discrepancies between MITF function in melanoma may be explained by the temporal differences in the analysis. For example, MITF levels are upregulated when cells or patients undergo MAPK inhibition treatment, resulting in a BRAF inhibitor drug-tolerant state, while MITF levels are decreased in progressed patients, unresponsive to therapy [69,77]. Recently, it has also been determined that these discrepancies in MITF function may be dependent upon the interaction between the transcription factors POU Class 3 Homeobox 2 (BRN2) and Paired Box 3 (PAX3) [77]. Smith and colleagues demonstrate that BRAF employs a PAX3/BRN2 rheostat to regulate MITF expression [77]. Under these conditions BRAF induces BRN2 expression to limit constitutive PAX3-induced transcription of MITF. This results in the downregulation of MITF expression and prevents MITF inhibiting BRAF-mediated proliferation. This crucial balance in MITF regulation further supports the hypothesis that MITF plays different physiological roles in different cellular contexts, ranging from differentiation, proliferation, or invasion. 

Also, it has clearly been demonstrated that MAPK inhibition leads to MITF upregulation, which itself, increases the levels of the OXPHOS related gene, PGC1α [68,78]. As such, MITF upregulation, following MAPK inhibition, appears to function through an adaptive response, leading to a micro-environment, which can support the outgrowth of a sub-population of melanoma cells. Approximately 10% of melanoma patients have focal amplifications at chromosome 3p13-3p14 [63]. Interestingly, both MITF and the lncRNA SAMMSON reside within this amplicon, suggesting that a proportion of melanoma patients alter oxidative metabolism through two unique independent mechanisms, the SAMMSON-p32 axis and the MITF-PGC1α axis, both of which are regulated by SOX10 [63]. 

Overall, MITF has been shown to be upregulated in response to MAPK inhibitors and, therefore, should be considered as a therapeutic target in combination with MAPK inhibitors. However, at this stage, the context-specific activity of MITF needs to be further delineated prior to specific patient populations being defined who may benefit from any proposed treatment. 

### 4.4. FBW7/USP28

Besides phosphorylation, a number of adaptive responses have been shown to be regulated by the activity of ubiquitin modifying enzymes, driving signalling in a number of oncogenic pathways [79,80]. Recently, it has been demonstrated that vemurafenib treatment alters the expression of a number of deubiquitinating enzymes (DUBs), including by increasing the levels of Ubiquitin specific protease 28 (USP28) [81]. Under these conditions, USP28 binds the F-box WD repeat containing protein 7 (FBW7) ubiquitin ligase complex, targeting all the RAF family members, including mutant BRAF^V600E^, for ubiquitin-mediated proteasomal degradation. FBW7 acts as a substrate recognition subunit, binding to consensus CDC4 phosphodegron (CPD) phosphorylation sites on BRAF, thereby permitting the recruitment of the SKP1-CUL1-F-box protein (SCF) ligase complex [82]. Furthermore, to maintain unwanted targeted degradation, FBW7 is auto-catalytically ubiquitinated and degraded, a process which is reversed in the presence of USP28 [83]. Therefore, FBW7 is able to bind and degrade BRAF^V600E^, while USP28 de-ubiquitinates and stabilises FBW7 [81]. Interestingly, it was shown earlier that SEL10 in *C. elegans*, which is the ortholog of FBW7 in humans, recognizes and binds to Lin45, a BRAF ortholog, through similarly conserved CPD phospho sites [84].

Furthermore, FBW7 protein levels is under the control of MAPK-mediated phosphorylation in pancreatic cancer. MAPK is able to phosphorylate FBW7 and target this protein for proteasomal degradation [85]. These results suggest the existence of a positive feedback loop embedded in MAPK signalling pathway, which plays a critical role in enhancing downregulation of MAPK signalling, following treatment with MAPK pathway inhibitors. Therefore, the stabilisation of the USP28/FBXW7 complex occurs through two independent mechanisms; the loss of ERK-mediated degradation of FBW7; and USP28 upregulation, resulting in deubiquitinating and stabilisation of FBW7, with the overall result being RAF degradation. FBXW7 was also discovered as one of the modulators of MITF and SOX10 protein level. It was shown that downregulation of FBXW7 in melanoma leads to increased MITF-mediated dysregulation of oxidative phosphorylation and increased SOX10 mediated migration of melanoma cells [86,87]. 

FBW7 and USP28 mutations have recently been identified in melanoma. Whole exome sequencing in a cohort of 77 melanoma samples, showing a number of recurrent mutations within the WD40 domain of FBW7, the binding domain targeting the CPD motif in FBW7 substrates, including RAF family members [88]. Similarly, USP28 was shown to be deleted in approximately 5% of all melanomas [81]. Furthermore, FBW7 inactivating mutations in colorectal cancer cells, caused the resistance to MAPK inhibitor therapies using Regorafenib and Sorafenib, and patients exhibiting low levels of USP28 displayed a shorter time to progression when treated BRAF/MEK inhibitor combination therapies [81,89]. These results highlight the importance of inactivation of FBW7/USP28 complex in BRAF^V600E^ mutated melanoma. Interestingly, a chemical compound screen identified the Polo Like Kinase 1 (PLK1) inhibitor and RAS mimetic, Rigosertib, as being synthetically lethal with USP28 downregulation in melanoma cells [81]. Recently, it has been demonstrated that CRAF associates with Aurora-A and PLKL1 at the centrosomes and its activity is required for mitotic progression. As loss of FBW7/USP28 enhances the expression of all three RAF family members, melanoma cells harbouring mutations in this complex may require the activity of both BRAF and CRAF for tumour progression. In line with these results, treatment of USP28 depleted cell lines, with Rigosertib-enhanced apoptosis, but did not alter the overall levels of phosphorylated ERK, nor the levels of the proapoptotic protein BCL2 Like 11 (BIM) [81]. Thus, suggesting that Rigosertib may be inducing apoptosis through MAPK-independent mechanisms. As USP28-mediated CRAF stabilization may play a role in cell cycle progression, combination therapies, targeting the MAPK pathway and the G2/M mitotic spindle assembly checkpoint, may be pursued as a potential viable treatment option in patients harbouring mutations in FBW7/USP28 complex. 

### 4.5. Autophagy

Tumours, which are under micro-environmental stress, such as starvation, preserve their organelles’ function and promote their own progression through a process called ‘autophagy’. Autophagy in cancer cells plays a crucial role in eliminating the dysfunctional mitochondria, while maintaining the normal function of mitochondria-mediated metabolism [90]. Therefore, autophagy has been introduced as a hallmark in some types of cancers, by which their progression is markedly dependent on this process. A role for autophagy in promoting tumorigenesis, in various BRAF mutant mouse models, has been supported. Xie and colleagues have demonstrated that ablation of *Atg7* in a *BRAF^V600E^/PTEN^null^* melanoma model decreased tumour growth, promoted senescence, and increased survival in comparison to mice that were wildptype for *Atg7* [91]. A similar approach has been used to assess the role of autophagy in lung cancer. The depletion of *Atg7* reduced late-stage *BRAF^V600E^*, and induced lung tumour burden, leading to increased overall survival in animal models [92]. Under these conditions it has been postulated that autophagy supplies the needed energy through continued glutamine metabolism to permit tumourigenesis. 

The inhibition of MAPK pathway leads to reduced glycolytic and mitochondrial function, while increasing the transcription of autophagy-associated genes, including the autophagy marker, LCS [43,93]. Furthermore, it was recently demonstrated that BRAF inhibition enhances autophagy through a transcriptional program, mediated by a Transcription Factor EB (TFEB)-TGFβ axis. Under these conditions BRAF inhibition suppresses ERK-mediated phosphorylation of the transcription factor TFEB, permitting nuclear localization and upregulation of TFEB target genes [94]. Also, it has been re-emphasized that one of the potential reasons for the ineffectiveness of MAPK inhibitors, in cancers with hyper-activation of the MAPK pathway, is the induction of autophagy, through the upregulation of LKB1-AMPK-ULK1 signalling axis [95]. Moreover, ERK inhibition enhances phosphorylated AMPK and Beclin1, aiding in the initiation of autophagy and localization of autophagic proteins to phagophore [43,95]. Autophagy may, therefore, be considered as an adaptive response to MAPK inhibition in melanoma. Various inhibitors have been designed to inhibit autophagy, which mainly inactivates this pathway by de-acidifying lysosomes and preventing the fusion of lysosome with autophagosome [96,97]. Among them Hydroxychloroquine (HCQ) and Lys06 have been used to treat melanoma patients successfully, reducing the overall tumour burden in these patients [93,98,99]. Similarly, other studies have shown the promising combinational effect of BRAF inhibitors and chloroquine in BRAF-mutated cancers [93,100,101]. Importantly, treatment of melanoma patient-derived xenografts, harbouring NRAS mutations, with a combination of a MEK inhibitor and a autophagy inhibitor, also significantly reduced the tumor burden [95]. Taken together, this effect highlights a potential Achilles heel in these difficult-to-treat cancers, with autophagic flux being an important mechanism of resistance to MAPK pathway inhibitors. However, Li and colleagues demonstrated that the inhibition of lysosome function or autophagy can un-expectantly augment the levels of TGFβ in the cells, resulting in increased Epithelial-Mesenchymal Transition (EMT) and metastasis [94]. This outcome suggests that a multi-pronged approach may be required to effectively target BRAF inhibitor induced autophagy. Finally, it is worth noting that, although, inhibition of autophagy in Atg7 knock out mice could lead to significant lung tumour reduction, it can also lead to infection and neurodegeneration, ultimately limiting survival in mice [102]. Therefore, the related toxicities for these types of therapies should be reasonably taken into consideration. 

## 5. Conclusions

The successful targeting of the MAPK pathway in melanoma has revolutionized the treatment of this disease. However, despite the success of these therapies, the duration of clinical response continues to be limited by primary or acquired resistance. In the majority of these cases, resistance is mediated by the presence of existing genetic lesions in other components of the MAPK pathway, resulting in the continued activation of the pathway, regardless of MAPK inhibitor treatment. Under these scenarios, it is likely that these mechanisms of resistance existed prior to drug exposure and are, therefore, selected out during treatment. In contrast, resistance, mediated by adaptive responses or feedback loops, likely allows cells to persist in an intermediate or adapted drug-tolerant state [17,42]. In regards to MAPK pathway re-activation, the relief of these negative feedback loops, following pathway inhibition, can never fully recapitulate the full oncogenic activity observed with BRAF^V600E^ [81]. Nevertheless, these drug-tolerant cells are capable of surviving initial drug therapy by exploiting the mechanisms highlighted above, by downregulating apoptosis or upregulating oxidative phosphorylation, proliferation, and autophagy. Importantly, however, is the potential ability of these drug-tolerant cells to survive and eventually evolve over time to acquire validated genetic resistance mechanisms. Therefore, targeting this reservoir of tolerant cells, prior to the development of full-blown genetic resistance, may offer a unique and important therapeutic opportunity. As such, the concomitant interrogation of patient-derived organoids, or the sequential biopsies of patients following treatment, may be required to fully understand patient specific adaptive responses, prior to implementing appropriate treatment strategies. 

## Figures and Tables

**Figure 1 cancers-11-01176-f001:**
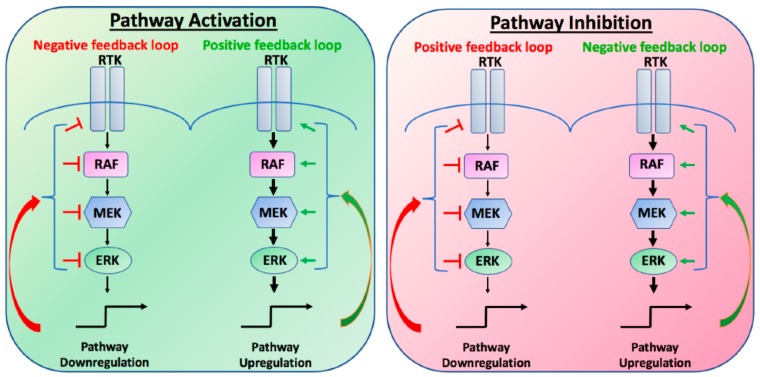
Schematic of positive and negative feedback loops in the canonical Mitogen-Activated Protein Kinase (MAPK) pathway. RTK, receptor tyrosine kinase; RAF, RAF proto oncogene; MEK, mitogen-activated protein kinase; ERK, extracellular signal-related kinase.

**Figure 2 cancers-11-01176-f002:**
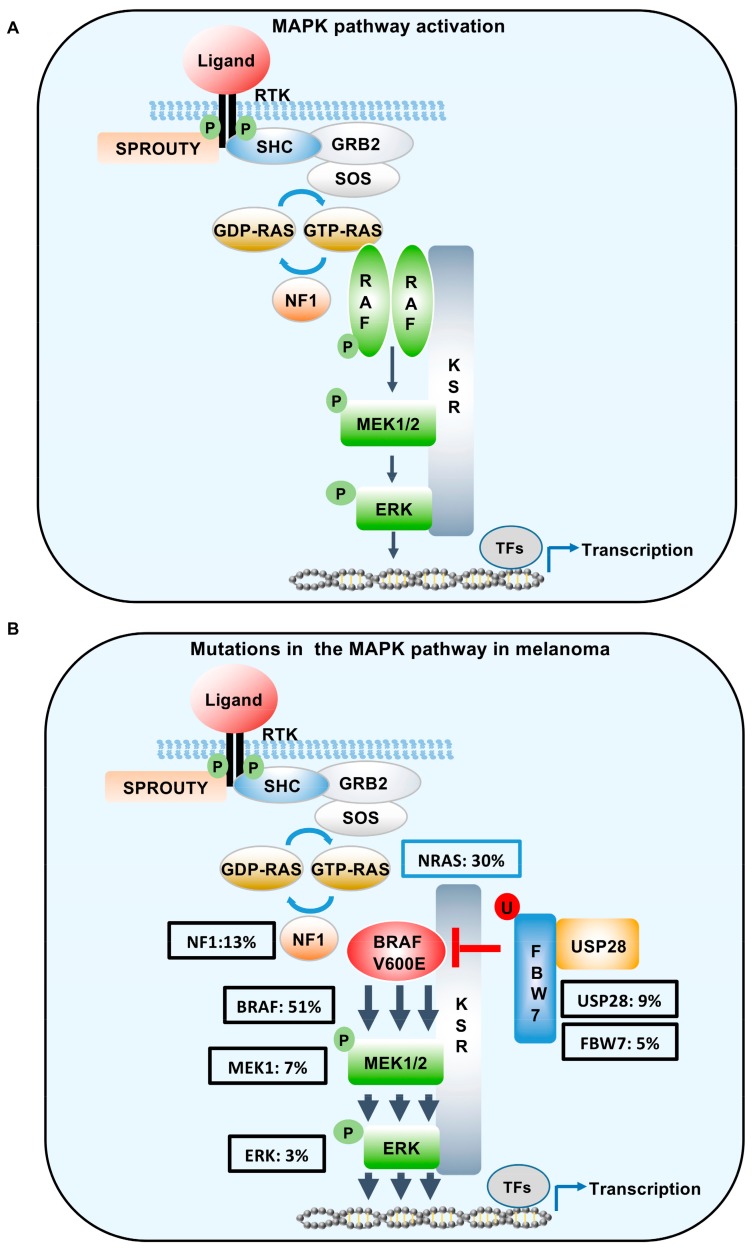
Schematic of MAPK signalling in the context of wild type BRAF and mutant BRAF^V600E^. (**A**) In cells expressing wild type BRAF, ligand-induced activation of RTKs enhances RAS-GTP activity resulting in the dimerization and activation of RAF family members. This binding serves to phosphorylate MEK1/2, which then phosphorylates ERK, resulting in ERK-mediated transcription. (**B**) Hyper-activation of the pathway in melanoma can occur through oncogenic mutations in NRAS, BRAF, MEK1, ERK, or the loss of function mutations in NF1, FBW7, and USP28. Class I BRAF mutations (V600E) function as monomers to engage the MEK/ERK signal cascade. Mutation statistics were extrapolated from cBioPortal [30]. SHC, SHC adaptor protein 1; GRB2, growth factor receptor bound protein 2; SOS, son of sevenless; RAS, RAS proto-oncogene; NF1, neurofibromatosis 1; FBW7, F-box and WD repeat domain containing 7; USP28, Ubiquitin specific protease 28; TF, transcription factors; MAPK: Mitogen-Activated Protein Kinase; BRAF: v-raf murine sarcoma viral oncogene homolog B; RTK: receptor tyrosine kinase; RAF: proto oncogene; MEK: mitogen-activated protein kinase; ERK: extracellular signal-related kinase.

**Figure 3 cancers-11-01176-f003:**
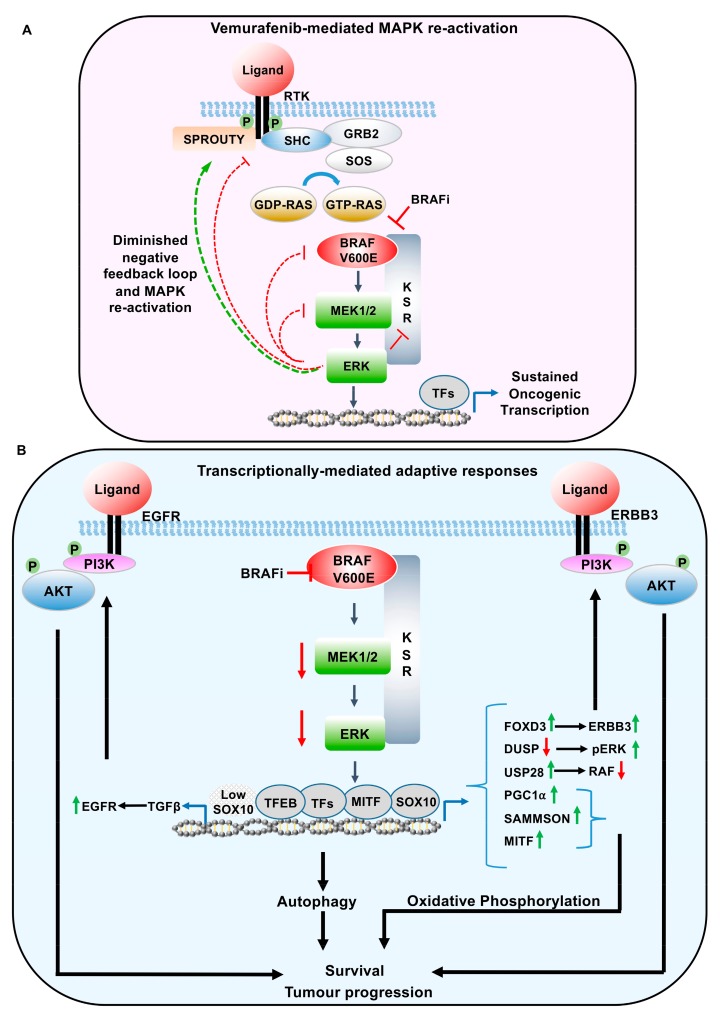
Adaptive response-mediated resistance to BRAF inhibitors in melanoma. (**A**) Negative feedback regulation of activated ERK on EGFR, RAF, and MEK, resulting in the downregulation of MAPK signalling. Figure 3 highlights the relief of these downstream kinase-dependent negative feedback loops, following MAPK inhibition. (**B**) Transcriptionally-mediated feedback loops regulating resistance to BRAF inhibitors. These feedback loops result in re-activation of the MAPK pathway (DUSP, USP28, FOXD3/ERBB3. TGFB/EGFR), parallel activation of PI3K pathway (FOXD3/ERBB3, TGFB/EGFR), increase oxidative phosphorylation (PGC1α, lncRNA SAMMSON, MITF), and autophagy related genes mediated by TFEB. RAF, RAF proto oncogene; MEK/MAPK, mitogen-activated protein kinase; ERK, extracellular signal-related kinase. SHC, SHC adaptor protein 1; GRB2, growth factor receptor bound protein 2; SOS, son of sevenless; RAS, RAS proto-oncogene; USP28, Ubiquitin specific protease 28; KSR, Kinase Suppressor of RAS; EGFR, epidermal growth factor receptor; ERBB3, Erb-B2 receptor tyrosine kinase 3; PI3K, Phosphoinositide 3-kinase; AKT, AKT serine/threonine kinase; MITF, melanocyte inducing transcription factor; SOX10, SRY-box 10; TGFβ, transforming growth factor β, TFEB, transcription factor EB; FOXD3, forkhead box D3; DUSP, dual specificity phosphatase; PGC1α, Peroxisome proliferator-activated receptor-gamma coactivator (PGC)-1alpha, TF, transcription factors.

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
