# Peer review of "Adaptive Responses as Mechanisms of Resistance to BRAF Inhibitors in Melanoma"

_cancers, 2019, doi:10.3390/cancers11081176_

Round 1

Reviewer 1 Report

Major comments:

1.     The manuscript needs significant and systemic editing. There are innumerous instances of sentences either phrased or punctuated poorly, typos and incorrectly used words, and inconsistency in capitalization when referring to gene/protein names (ex. RAS vs Ras). Some of the more egregious examples include:

·      Lines 84-86, 142-144, require re-writing, due to either illegibility or poor syntax leading to inaccurate statements.

·      Lines 49-61 are written in a confusing manner, specifically with relation to explaining the concept of positive feedback in pathway inhibition. Example: “inhibition of the pathway may be further downregulated” does not make sense. If the section pertaining to feedback loops in pathway inhibition is to be retained, then a clarification of the technical definitions of positive and negative feedback should be included. There is also a typo in the figure title.

·      Lines 237-243 appear to be composed of entirely unsubstantiated speculation. This section requires supporting data. 

2.     This review needs to be clearer in its focus on intrinsic adaptive responses, and what fits into that category. Section 3 covers mutations within the MAPK pathway, which are intrinsic but not adaptive, while ignoring numerous other mutations which are not in MAPK components but still contribute to BRAFi resistance. Section 4 begins by covering all rates of inherent resistance to vemurafenib (much of which are caused by intrinsic mutations), and then ignores those to briefly talk about paradoxical activation, before finally moving to feedback inhibition. There is no means proffered as to how most of feedback inhibition mechanisms could be dysregulated adaptively. 

3.   Section 5 introduces autophagy, but provides no data to show that it is adaptive in the melanoma context. 

4. The manuscript overall seems meandering in focus, and needs to decide whether it is going to cover all intrinsic causes of BRAFi resistance, or only the intrinsic adaptive ones. In either case, there are significant manuscript revisions required, to either excise the portions covering non-adaptive mechanisms, or cover the many intrinsic mechanisms not currently included.

5.     The final paragraph of section 5 appears tacked-on, and the lack of mechanistic characterization means it does not fit well with the theme of the section (transcriptionally-mediated feedback loops). It also almost completely ignores data from melanomas in favor of discussing autophagy in other cancers, which is baffling as there is melanoma data available. 

6.     The conclusions section is not actually a conclusion, but a summary of current therapy and speculation regarding future therapies which have no relation to anything discussed in this manuscript.

Minor comments:

1.     Figures 2A-B and 3A-B are essentially identical, and 3A-B is in fact only a minor change that does not adequately make its point unless being directly compared to 2A-B. These would benefit from being reworked, and probably combined into a single figure.

Reviewer 2 Report

This review article provided some new ideas and insightful concepts into the BRAF/MAPK inhibition field.  However, there are three major weakness of this article: 

the writing is difficult to follow, with many, many run-on sentences and unclear meaning.  A few examples: a) line 182-line 185: "Furthermore, these intrinsic adaptive responses leading to resistance can be divided into non-transcriptional based adaptive responses resulting in the regulation of post-translational  modifications of upstream kinases, phosphatases and ubiquitin modifying enzymes, or 185 transcriptionally mediated adaptive responses."  b) Line 194-196: "Now termed the “RAF inhibitor paradox” this area of research has been under intense investigation and as a consequence a deeper understanding of RAF dimerization and activation has partially clarified this phenomenon"

some basic concepts are wrong: e.g., Line 46-47: "However, little has been recognised on the intrinsic adaptive responses that are initiated 47 following BRAF inhibition that limit BRAF inhibitor therapy in melanoma."   This is not true.  There have been numerous reviews and studies on this topic.

The paper is poorly organized, which made it even harder to follow.

Minor issues: figures are of low resolution and difficult to read; figure legends are unclear or not sufficient to explain the figures.  

Reviewer 3 Report

Manuscript Title

 Intrinsic Adaptive Responses as Mechanisms of  Resistance to BRAF Inhibitors in Melanoma

 Azad Saei 1 and Pieter Johan Adam Eichhorn

Authors investigated MAPK signalling in the context of wild type BRAF and mutant BRAFV600E. melanoma cells. This topic participates to the crucial understanding of BRAF resistance and discloses several new approaches to treat human melanoma. We believe that this Review is well exostiv and may help several researchers involved in the in vitro/in vivo treatment of the different types of melanoma.

We believe that these following papers of literature have to be enclosed in the paper, also to introduce possible new aspects of melanoma resistance.

1: Laurenzana A, et al . EGFR/uPAR interaction as druggable target to overcome vemurafenib acquired resistance in melanoma cells. EBioMedicine. 2019 Jan;39:194-206.

2: Ruzzolini J, et al . Everolimus selectively targets vemurafenib resistant BRAF(V600E) melanoma cells adapted to low pH. Cancer Lett. 2017 Nov 1;408:43-54.

Round 2

Reviewer 1 Report

I am happy with the response to my suggestions.

Author Response

We thank the reviewer for taking the time to read our manuscript.

Reviewer 2 Report

This is a much improved version.  Minor problems include missing citations and some apparent errors in texts.  English still need to be improved. 

Example of missing citations:

1) line 321-326. need citations.

2) miss-cited: reference 69

Other references are not checked by this reviewer but I think citations need a thorough check.

Concept errors in texts: (only an example) 

Line 329-331, MART1 and GP100 are NOT OXPHOS-related genes but the way this sentence is written made it sound like so.

Author Response

Cancers-552807

Adaptive Responses as Mechanisms of Resistance to BRAF inhibitors in Melanoma.

Responses to reviewer’s comments:

We thank the reviewers for the thoughtful and thorough review of the manuscript. We have revised the manuscript according to the reviewers’ suggestions and have addressed all the concerns brought forward by the reviewers.

Reviewer#2: Minor problems include missing citations and some apparent errors in texts.  Line 321-326, need citations. Miss-cited: reference 69. Other references are not checked by this reviewer but I think citations need a thorough check.

We thank the reviewer for their thorough revision of our manuscript. Thanks to their comments we noticed that a number of references were incorrect. We have now edited these references where applicable.

Reviewer#2: Line 329-331, MART1 and GP100 are NOT OXPHOS-related genes but the way this sentence is written made it sound like so.

We thank the reviewer on noticing this error on our part. As this paragraph is focused on oxidative phosphorylation we have removed MITF regulation of MART1 and GP100. As the reviewer correctly pointed out they are not involved in oxidative phosphorylation.

Reviewer#2: English still need to be improved. 

 We have gone over our manuscript and attempted to correct any grammatical errors or potential run on sentences we identified.